# Managing Virtual Presenteeism during the COVID-19 Pandemic: A Multilevel Study on Managers’ Stress Management Competencies to Foster Functional Presenteeism

**DOI:** 10.3390/ijerph21091115

**Published:** 2024-08-23

**Authors:** Sandra Salvoni, Caroline Biron, Marie-Hélène Gilbert, Julie Dextras-Gauthier, Hans Ivers

**Affiliations:** 1Department of Management, Faculty of Business & Administration, Université Laval, Québec, QC G1V 0A6, Canada; caroline.biron@fsa.ulaval.ca (C.B.); marie-helene.gilbert@fsa.ulaval.ca (M.-H.G.); julie.dextras-gauthier@fsa.ulaval.ca (J.D.-G.); 2VITAM—Research Center for Sustainable Health, Université Laval, Québec, QC G1J 2G1, Canada; 3School of Psychology, Université Laval, Québec, QC G1V 0A6, Canada; hans.ivers@psy.ulaval.ca

**Keywords:** virtual presenteeism, telework, mental health, job performance, stress management competencies, health-performance framework of presenteeism, COVID-19

## Abstract

Teleworking remains an attractive option for many workers since the COVID-19 pandemic, but it presents significant management challenges, particularly when employees face health issues. The management of virtual presenteeism, where employees continue teleworking despite being ill, has received limited attention. This study explores the relationship between managers’ stress management competencies (SMCs), mental health, and job performance of virtual presentees, aiming to fostering more functional presenteeism. We examine whether managers’ SMCs promote functional presenteeism by comparing managers’ self-assessments with employee assessments, and analyzing how agreement levels between the two affect mental health and job performance. Data were collected from 365 teleworkers supervised by 157 managers in a large public organization in Québec. The results indicate that virtual presentees’ mental health and job performance are closely linked to employees’ assessment of their managers’ SMCs. Employees who agreed with their manager or overestimated their managers’ SMCs exhibited better mental health and job performance than those who agreed with their manager on low SMCs or underestimated their managers. This study expands on the health-performance framework of presenteeism and self-other agreements, highlighting management practices that should be enhanced in the context of virtual presenteeism.

## 1. Introduction

The COVID-19 pandemic has significantly changed the landscape of work, with a large number of workers transitioning from traditional office settings to remote telework. The percentage of Canadians completing most of their working hours in a given week from home rose to about 40% in April 2020 but fell to about 20% in November 2023 [1]. A comparable trend emerged in Québec, where by 2022 a significant portion of the workforce (34.5%), especially managers, professionals and high-income earners are now engaged in telework [2]. Almost 40% of these teleworkers work from home exclusively, and 27% for more than half of their hours.

However, despite the advantages of working from home, such as increased autonomy, improved work-life balance, flexibility, and the reduction in commuting time leading to increased productivity and higher job satisfaction [3], working predominantly or entirely remotely may have negative effects, particularly concerning “presenteeism” [4,5,6,7]. Presenteeism is the behavior of working while being physically or mentally unwell [8,9]. Telework allows workers to adjust their work conditions at home, and thus is associated with the perception that it is less demanding than working onsite, especially when managing health issues [7]. In a work-from-home environment, this perception leads to the behavior of continuing to work, even when unwell, which is called “virtual presenteeism”, as identified by Ruhle and Schmoll [10].

Presenteeism is a rising concern for managers due to its direct impact on health and performance [11]. According to the health-performance framework of presenteeism (HPFP), presenteeism can be functional when the workplace offers support and adequate resources for adaptation, such as strong management competencies. However, presenteeism is often thought of as dysfunctional, characterized by behavior that harms health and performance, frequently leading to absenteeism and a decline in future well-being due to overextension, inadequate recovery, and poor management. In such cases, productivity demands can take precedence, overstretching work capacity and hindering recovery [12]. Presenteeism can also have a therapeutic aspect, where the emphasis is more on health recovery and less on performance demands, allowing virtual presentees to recuperate while working from home. Another form, over-achieving presenteeism, occurs when an individual maintains a high level of performance but at the expense of their recovery from illness. Ultimately, the goal is to achieve functional presenteeism. According to the health-performance framework of presenteeism (HPFP) [12], when workplaces provide adequate support and resources for adaptation—such as effective management competencies—presenteeism can be functional. Functional presenteeism allows employees to recover from health issues (or maintain relatively good health) while sustaining strong work performance [12].

Virtual presenteeism can pose a greater challenge because managers do not have face-to-face or frequent access to their employees, and may thus be less familiar with them. Consequently, it may be harder to detect health issues among employees whose work effort and productivity are likewise less visible to colleagues and managers [13]. This challenge is further exacerbated when employees conceal illness symptoms or feign being healthy, a phenomenon known as “sickness surface acting” to maintain a positive appearance while working from home while ill [14]. According to Dewa, van Weeghel [15], employees require good relationships with their managers to feel comfortable disclosing mental health issues.

Since the onset of the pandemic, the increased prevalence of mental health issues means that these are now more likely than physical health problems to increase presenteeism, which poses a specific challenge to presenteeism research [16]. Due in part to the difficulty of disclosing and recognizing mental health issues at work, related instances of presenteeism often go unrecognized. Indeed, the stigma surrounding psychological and emotional issues causes some individuals to consider related absences as less valid than those caused by visible physical health issues, which may contribute to a persistent pattern of presenteeism [17] as they may hesitate to take time off because they think they will be invalidated. Thus, presenteeism related to mental health issues may be underestimated or inadequately addressed in many studies. This gap has been recognized by the World Health Organization (WHO) and the International Labour Organization, who have called for measures to safeguard teleworkers’ health [18].

Answering early calls to promote and facilitate more functional presenteeism, namely a return to optimal health and performance, is a significant concern for organizational management [19], thus it is important to understand how to better manage virtual presenteeism. Once managers are aware of these health issues, they may face challenges to determine the most appropriate behaviors to promote functional presenteeism and prevent further deterioration of mental health and job performance.

Often, the only readily available strategy is to suggest sick leave (absenteeism) for those who are unwell [6]. The Chartered Institute of Personnel and Development [6] report indicates that 47% of employers grappling with presenteeism are not implementing corrective or preventive measures. The Stress Management Competencies (SMCs) are particularly important, as emphasized by Ferreira, Mach [20], who highlighted the need to study measures and develop interventions to address virtual presenteeism. Such SMCs are more accurate predictors of well-being and overall health than broader factors such as leadership style [21]. Existing leadership models do not comprehensively address manager behaviors relevant to managing work-related stress [22], and it is recommended to investigate specific health-related leader behaviors [23] and use more specific competency predictors related to health, rather than a broad leadership model.

Managers influence employees’ mental health through their stress management competencies (SMCs)—the tangible behaviors they exhibit [24,25,26]. Given the challenges of detecting and managing mental health issues in telework, the SMCs framework is particularly valuable. It highlights specific management behaviors that can help reduce exposure to stressors. However, this framework has not yet been applied to virtual presentees’ mental health.

This study advances knowledge on the management of presenteeism by bridging three gaps. First, it addresses the lack of research on managing virtual presenteeism to promote functional behavior. Second, it contributes empirically to the HPFP literature by examining SMCs that support functional presenteeism in teleworking settings. Lastly, it extends the self-other agreement (SOA) literature to foster functional presenteeism by exploring the perceptual gap between employees and managers, revealing valuable insights for practitioner and manager competency development.

### 1.1. Stress Management Competencies (SMC)

Managerial behaviors are linked to employee presenteeism [27] and are significantly associated with mental health [28]. However, the specific behaviors of managers that enhance employee functional presenteeism remain unclear, especially in the context of telework. The SMC Indicator Tool (SMCIT) [29] provides a framework of specific management behaviors designed for work stress and well-being. This tool is particularly suitable for this study as it provides a framework of management behaviors designed for work stress and well-being. Toderi and Sarchielli [30] revealed that, while each of the four SMCs studied by the SMCIT was present in different leadership frameworks, none encompassed all four SMCs.

The SMCIT assesses four main SMCs: being respectful and responsible (RR; integrity, managing emotions, and a considerate approach), managing and communicating existing and future work (MCW; proactive work management, problem-solving, and participative/empowering), reasoning and managing difficult situations (RDS; managing conflicts, using organizational resources, and taking responsibility for resolving issues, and managing the individual within the team (MIT; personally accessible, sociable, and empathetic engagement).

The validation of the manager’s questionnaire showed that the factor structure found in employee responses was also applicable to manager data [29]. Specifically, self-assessed SMCs were significantly and positively related to employee-attributed SMCs. The structure of both SMCIT versions is the same, which facilitates comparisons of manager self-assessments and employee perceptions.

### 1.2. The Health-Performance Framework of Presenteeism (HPFP)

The literature reveals both adverse and favorable outcomes associated with individuals choosing to work while ill. Karanika-Murray and Biron [12] framed presenteeism as a dynamic behavior focused on the individual, and have proposed the HPFP, which views it as part of a decision-making process where individuals maintain performance despite suboptimal health, which confers adaptive benefits and serves individuals by reinforcing their sense of responsibility, identity, and values [8]. Under certain circumstances, presenteeism can be a healthy and sustainable choice for some individuals [17], which supports the idea of positive presenteeism [25].

While most research to date has focused on the negative aspects of presenteeism, the HPFP model offers a more nuanced perspective by identifying four distinct profiles of presenteeism, each with varying degrees of impact on health and job performance. The HPFP model uses a 2 × 2 typology to classify these profiles based on the severity of health and performance impairments: a functional profile (good health, good performance), a dysfunctional profile (poor health, poor performance), a therapeutic profile (poor performance despite benign health issues), and an overachieving profile (high performance at the expense of health). Notably, these profiles have not yet been explored within a virtual work context. 

Based on the conservation of resources theory [31,32], Karanika-Murray and Biron [12] suggest that functional presenteeism—employees working and meeting performance expectations while unwell without worsening their health—may help balance health constraints with performance demands, perhaps even preventing health or productivity declines. However, despite the concept advancing our understanding of presenteeism behavior, no guidelines currently exist to promote functional presenteeism (enhanced mental health and job performance) as a means of returning to optimal health or avoiding more harmful types of presenteeism. Fostering functional presenteeism should be advantageous for both individuals and organizations. However, it requires offering employees options for adjusting their work tasks according to health-related capacity [33]. For managers, fostering functional presenteeism means adjusting behaviors and attitudes that promote both a return to health and satisfactory work performance by their employees. This can represent an additional challenge when managing employees who telework and have mental health problems.

As presenteeism is a person-centered phenomenon [12] and requires a person-centered approach, both axes of the HPFP framework variable (health and performance) should be studied at the individual level. The first axis of the HPFP framework is health. Given the significant mental health challenges that emerged during the COVID-19 pandemic—where poor mental health in Canada reached unprecedented levels and Québec saw a sharp decline in mental health quality from 67% to 58% between 2019 and 2022 [24]—our study focuses specifically on improving the mental health of virtual presentees rather than global health. This focus is particularly crucial given the critical role mental health plays in the context of presenteeism. Traditionally viewed as unidimensional, mental health is now understood as bidimensional [34], encompassing both psychological well-being (PWB) as the positive dimension and psychological distress (PD) as the negative dimension. This negative dimension, represented by psychological distress (PD, is especially relevant when examining presenteeism and its relation to health impairment. By concentrating on mental health, our study aims to provide deeper insights into how stress management competencies can be related to the mental health and job performance of employees, especially in the context of virtual presenteeism.

The second axis of the HPFP framework is job performance, which we examine at the individual level, consistent with the person-centered approach. This approach focuses on specific actions and behaviors that individuals engage in, which directly contribute to achieving the organization’s goals [35]. The primary goals of individual work performance are to assess overall employee status and identify changes in individual work performance, such as those resulting from worksite health promotion interventions [36]. Recent literature has made several efforts to model the substantive content domains of individual performance [35]. Despite the fact there is a consensus regarding the multidimensional nature of performance [37] where three major domains of job performance were identified [38] (task performance, contextual performance, and counter-productive work behavior), it offers a comprehensive and concise approach for evaluating overall job performance [37], and the application of these various performance dimensions raises issues and introduces biases [39,40]. In an organizational and applied context, it can be legitimate to use a unidimensional conceptualization of job performance, particularly using the HFPF [41], as performance is represented on a single axis in the model. The unidimensional individual job performance is the choice made in “Section 2”.

### 1.3. Hypotheses Development

In the context of COVID-19 and teleworking conditions, relationships with immediate managers have changed, and the risk of social isolation has increased, especially in the lockdown period. It is important to recognize that effective management extends beyond mere actions and also encompasses how these actions are perceived and understood by employees [42]. Toderi and Balducci [29] found that employees who perceived their managers as competent tended to experience more positive emotions and fewer negative ones. This finding is likely applicable to the phenomenon of virtual presenteeism, suggesting that when employees recognize managerial competence, it could positively impact their mental health and job performance.

Considering that managers often face challenges in managing presenteeism within a telework context, and acknowledging that self-assessments of leadership skills are not sufficient predictors of management and leadership effectiveness [43,44], we deemed it essential to include external evaluations of the managers’ SMCs for a more accurate measurement.

**H1a:** 
*Employee perceptions of their managers’ stress management competencies are negatively related to employee psychological distress.*


**H1b:** 
*Employee perceptions of their managers’ stress management competencies are positively related to employee job performance.*


Notably, employee assessment indicate that managers often overestimate their own competencies [42]. Managers often struggle to align their actual practices with their intentions for various reasons, including time constraints, resistance, and limited resources [45]. Perceptions of the manager’s intentions often differ significantly among the manager and their team [46], nevertheless, manager self-assessment offers valuable insights into how managers perceive their abilities and behaviors. Intended, actual, and perceived practices are distinct yet interconnected concepts [45]. Even if managers’ self-assessments are different from those of employees, we hypothesize that the relationship between positive manager self-assessment and teleworkers’ mental health and performance must be positive:

**H2a:** 
*Manager self-assessments of their stress management competencies are negatively related to employee psychological distress.*


**H2b:** 
*Manager self-assessments of their stress management competencies are positively related to employee job performance.*


The discrepancy between perceptions, poor communication, or ambiguous behaviors may lead to differences of opinion [47], while closer interactions between managers and employees may help align perceptions. Consequently, we chose to explore the alignment between managers’ self-assessments of their stress management competencies (SMCs) and their employees’ perceptions of these same competencies. In teleworking environments, the lack of social interaction can lead to difficulties in interpreting cues, influencing how employees perceive managerial behavior [13,48]. This situation may result in employees’ SMCs being different, or employees being unaware of their managers’ intentions. Thus, we hypothesize that even if perceptions differ, due to their common basis in actual SMCs, manager-intended, and employee-perceived SMCs should be positively associated, even if the relationship is weak [46].

**H3:** *Managers’ self-assessments of their stress management competencies are weakly and positively correlated with employee perceptions of their managers’ SMCs*.

Research on self-other agreement (SOA) between managers’ and employees’ perceptions is well established, however, its application to stress management competencies or healthy leadership, particularly in the context of virtual presenteeism, has been limited. This study addresses this gap, emphasizing that SOA has not yet been thoroughly explored in relation to these factors and their potential impact on promoting functional virtual presenteeism. Studies consistently reveal disparities between managers and their teams in how managers’ intended behaviors are perceived (e.g., [46]). Building on this, our study examines the concept of congruence between managers’ and employees’ perceptions of managers’ SMCs, as outlined by Fleenor, Smither [43], to explore how these perception gaps may be linked to mental health and job performance outcomes of virtual presentees.

According to Tafvelin et al. [49], outcomes improve when both managers and their employees provide high and congruent assessment. Conversely, discrepancies between the two have been linked to lower employee health and work performance [43,50]. Lee and Carpenter [44] conducted a meta-analysis of mean differences between managers’ and employees’ assessments to determine whether managers or employees tend to over-report or under-report SMCs. Building on this model (see Table 1), we propose an SOA approach to understanding the relation with employees’ mental health and job performance. Furthermore, a recent study by Toderi et al. [51] highlighted that employees report a better psychosocial environment when supervised by managers who either align well with employee perceptions or understate their own competencies, whereas lower ratings are observed under managers who overestimate their competencies or who are perceived poorly by employees.

In this study, agreement (high) is achieved when a manager’s self-assessment of their stress management competencies (SMCs) is higher than the average self-assessment of other managers, and when employees’ perceptions of their managers’ SMCs are higher than the perceptions of other employees regarding their managers’ SMCs. Agreement (low) is achieved when a manager’s self-assessment of their stress management competencies (SMCs) is lower than the average self-assessment of other managers, and when employees’ perceptions of their managers’ SMCs are lower than the perceptions of other employees regarding their managers’ SMCs. Underestimate means: employees’ perceptions of their managers’ SMCs are lower than the average perceptions of other employees regarding their managers’ SMCs. Overestimate means: employees’ perceptions of their managers’ SMCs are higher than the average perceptions of other employees regarding their managers’ SMCs.

**H4a:** 
*High manager-employee agreement on the assessment of the manager’s stress management competencies is associated with lower psychological distress and higher job performance.*


**H4b:** 
*Low manager-employee agreement on the assessment of the manager’s stress management competencies is associated with higher psychological distress and lower job performance.*


**H4c:** 
*Employees who underestimate manager’s stress management competencies exhibit higher psychological distress and lower job performance.*


**H4d:** 
*Employees who overestimate manager’s stress management competencies exhibit lower psychological distress and higher job performance.*


## 2. Materials and Methods

### 2.1. Participants and Data Collection

After obtaining ethical approval (2020-081/23-03-2021), we used the Redcap platform to send an online questionnaire to 220 managers (out of 245 direct managers) within a large provincial public service organization in Quebec, dedicated to health and safety, comprising 65% professionals and 35% civil servants and blue-collar workers. At the same time, in February–April 2022, we solicited participants from a random selection of employees nested in the managers’ teams, which represented approximately 50% of the organization’s 4760 employees. Multisource data were gathered from various levels of the hierarchical structure during the study, which included 880 employees and 190 managers. We then applied multilevel analysis, with employees nested among managers, to examine the data. These sources provide different perspectives on the same phenomenon.

This study was conducted in the first quarter of 2022, a period when telecommuting was mandatory or strongly recommended due to the increase in Omicron variant cases. This situation presented unique challenges for managers in identifying employees who were teleworking while ill, as traditional in-person cues were absent. Since it is more difficult to identify those engaging in presenteeism, we aimed to study which managers’ stress management competencies could be linked to virtual presentees, regardless of whether the manager was aware of this behavior or not.

Employees who worked remotely for at least 80% of the time over the previous 7 days (N = 615) were included, as were their managers (N = 179). Employees who declared at least one day of presenteeism during the 28-day period were considered to have engaged in virtual presenteeism (N = 365). Of the included participants, 79.24% of the employees and 60.8% of the managers identified as women. The age distribution of the selected sample was as follows: 48.6% of employees and 21% of managers were between 25 and 45 years old, 34.1% of employees and 56% of managers were between 45 and 54 years old, and 17.3% of employees and 23% of managers were aged 55 or older. Regarding their occupations, 64.8% of the participants were professionals. The employee participants had, on average, 9.96 years of seniority, and the manager participants had, on average, 14.6 years of management experience. On average, the participants each worked 35 h per week (see Table 2).

### 2.2. Materials

#### 2.2.1. Telework

Participants were asked to indicate the number of hours worked from home and the number of hours worked onsite during the last week (seven days). A full-time teleworker is defined as an individual who spends at least 80% of their working hours in each data collection period working from home. The 80% threshold is determined based on the average number of hours teleworked relative to the total working hours completed over the past seven days [4].

#### 2.2.2. Presenteeism

Despite the common use of a 12-month recall period in presenteeism studies [17], the suitable time frame for evaluating presenteeism is not entirely clear [52]. Accordingly, based on the study by Navarro and al. [53], as well as Ruhle and Breitsohl [52]’s recommendations, we chose a shorter period to reduce the influence of recall bias and minimize internal validity threats. As recommended by Johns [54], we measured presenteeism using one open-ended question: “During the last month (28 days), how many days did you work while experiencing a health issue?” This question served as a continuous variable for presenteeism and, based on the responses, individuals who reported 0 days were not considered to have engaged in presenteeism and were thus excluded from the study.

#### 2.2.3. Mental Health

We measured mental health using the 6-item Kessler Psychological Distress Scale [55]. The scale’s items (e.g., “About how often during the last month did you feel nervous?”) are scored from 0 (none of the time) to 4 (all of the time).

#### 2.2.4. Job Performance

Overall employee job performance was measured using the Individual Work Performance Questionnaire (IWPQ; [36]). The IWPQ consists of 18 Likert-type items ranging from 1 (rarely) to 5 (always), referring to the last month: five task performance items (e.g., “I managed to plan my work so that I finished it on time”), eight contextual performance items (e.g., “I took on challenging tasks when these were available”), and five counterproductive work behavior (reversed) items (e.g., “I complained about unimportant issues at work”). This scale has a good reliability (α = 0.805). Individual work performance was considered as unidimensional considering the applied organizational context. This is corroborates Viswesvaran, Schmidt and Ones [39] and Santalla-Banderali and Alvarado [40], who raised several issues regarding the multi-dimensional conceptualization of work performance, and is conceptually consistent with the HFPF [39].

#### 2.2.5. Stress Management Competencies

The SMCIT-36 [30] measures four key competencies and behaviors on a 5-point Likert-type scale ranging from 1 (strongly disagree) to 5 (strongly agree). A 6-month recall period is specified by asking managers and employees to evaluate managers’ practices over the last 6 months. While leadership and management competency assessments typically do not specify a timeline, we chose this 6-month period to capture the most recent behaviors, given the unique context of the pandemic and teleworking. This timeframe was selected to provide employees with enough time to assess management competencies accurately, considering the limited contact with managers during the data collection period. Examples of items that measure each of the four competencies are as follows: “Doesn’t speak about team members behind their backs” (RR), “When necessary, will stop additional work being passed on to me” (MCW), “Deals objectively with employee conflict” (RDS), and “Returns my calls/emails promptly” (MIT). The employee SMCIT-36 assessment scale was adjusted by rephrasing statements for managerial self-assessment. To ensure the accuracy of the wording in the study context, two bilingual members of the research team, who are also experts in this field, back-translated both versions of the questionnaire into French. The study aligns with the findings of Toderi and Sarchielli [30] for the employee sample, as it exhibits good Cronbach’s alpha values ranging from α = 0.79 to 0.86 for the four subscales of SCMIT-36. As for the manager sample, the subscales exhibited similar Cronbach’s alpha values ranging from α = 0.76 to 0.85. However, the RR (respectful and responsible) subscale showed a notably low internal consistency with a Cronbach’s alpha of α = 0.35. This deviation is significant and does not align with the results reported by Toderi and Sarchielli [30].

#### 2.2.6. Control Variables

Following Miraglia and Johns [17], we controlled for the following variables related to presenteeism: gender (0 = male and 1 = female, with a positive effect) and age (younger individuals are more prone to presenteeism).

### 2.3. Analysis

A random intercept model was conducted to assess the variance attributable to teams and individuals, and to calculate intraclass correlation coefficients (ICC), with a higher ICC indicating greater variance at the team level. The obtained ICC values showed moderate to modest associations between manager SMCs and employee mental health and performance, which justifies a multilevel approach to statistical analysis (see Table 3).

We employed multilevel data analysis with employees (Level 1) nested among team managers (Level 2) and conducted multilevel logistic regression analysis (nested by managers) to estimate the odds ratios of the relationships between the SMCs most relevant to mental health and job performance in the virtual presenteeism context. Analyses were conducted using Statistical Analysis System (SAS) 9.4. The objectives were as follows: (1) explore the relationship between managers’ SMC self-assessments and employee perceptions of their managers’ SMCs on employee mental health and job performance; (2) understand the relationship between managers’ self-assessment of their own SMCs and the employee perceptions of their managers’ SMCs; and (3) explore the gap between SMCs, as reported by managers and perceived by employees, and its relationship to employee mental health and job performance.

## 3. Results

Table 4 presents the study variable correlations and descriptive statistics.

When teleworking presentees perceive their manager as competent (indicated by a higher SMC score), they report lower psychological distress (better mental health) and better job performance, thus, H1a and H1b are supported. However, the associations between manager self-assessments of SMCs and employee mental- and health-performance indicators were rather weak and mostly nonsignificant. In general, managers’ self-assessments of their SMCs were not significantly related to employee mental health and job performance. The only exception was the managers’ self-assessment of their “Managing and Communicating Existing and Future Work” competency, which was significantly associated with employee job performance. Thus, H2a and H2b were not fully supported (see Table 5).

H3 posited a weak correlation between managers’ self-assessments of SMCs and employees’ assessment of these competencies. Bivariate analyses were carried out to understand the relationship between managers’ self-assessed SMCs and employee perceptions (see Table 6).

Bivariate analyses revealed small correlations between manager-intended SMCs and employee-perceived SMCs, which highlighted a significant difference between these two evaluation sources. The findings also suggest a tendency among managers to overestimate their SMCs compared to employee perceptions, with managers self-assessing their SMCs on 
$\bar{x}$
 4.15 with σ = 0.26, and employees assessing them on 
$\bar{x}$
 = 3.98, with σ = 0.58.

The agreements and disagreements between managers and their employees were categorized into four groups based on (self-)assessment Z-scores. For managers, a score of less than 0 indicates a below-average self-assessment; for employees, a score below 0 indicates a below-average manager SMC self-assessment (see Table 7).

As anticipated, a combination of aligned and elevated assessments (Agree-High), along with instances where employee SMC perceptions were higher than manager self-assessments (Disagree-Over), was positively associated with performance and negatively associated with PD. Conversely, the combined association of aligned, low assessments (Agree-Low), with instances where employee SMC perceptions were lower than managers’ self-assessment (Disagree-Under), was, as expected, significantly weaker. In this context, “significantly” denotes a value exceeding one half of a standard deviation from the average self-other differences.

As expected, Agree-High assessments are associated with reduced PD and increased JP, while Agree-Low assessments will be detrimental to JP and increase PD. Contrary to expectations, for the RR SMCs, being in disagreement is not statistically different from being in agreement, but the level of assessment agreement is related to outcomes. For the RDS SMCs, as expected, the level of agreement or disagreement was not associated with PD but with performance. Finally, an employee who undervalues MIT competencies will not exhibit a significant difference in PD. In essence, positive employee perceptions reduce PD and enhance performance, regardless of manager self-assessments.

## 4. Discussion

This research explored how managers’ SMCs were linked to improved mental health and job performance among teleworking employees who reported engaging in presenteeism, thereby promoting more functional presenteeism as outlined in the HPFP framework. It also investigated the concept of congruence [43] between managers’ and employees’ perceptions of managers’ SMCs to explore the relationship between assessment gaps and virtual presentees’ mental health and job performance outcomes.

It employed the SMC framework [30] and HPFP model [12] to highlight managers’ SMCs that reduce PD, enhance JP and contribute to more functional presenteeism in virtual work settings, enhancing mental health without compromising job performance. The findings reveal a positive relationship between employees’ perceptions of SMCs, and their mental health and job performance. Regardless of the specific stress management competencies perceived by employees, all were positively linked to the mental health and job performance of virtual presentees. These behaviors should be valued and promoted among managers. In contrast, managers’ self-assessments of their competencies had a limited impact on mental health and job performance, with only the “Managing and Communicating Existing and Future Work” competency showing a positive correlation with employee job performance. We did observe a significant disparity between managers’ self-assessments and employees’ perceptions of manager’s SMCs, and it was the employees’ evaluations, not the managers’ self-assessments, that were most strongly correlated with functional presenteeism. These results emphasize that, to enhance functional presenteeism, the focus should be on employees’ perceptions of managers’ SMCs. These findings align with a study that utilized a multilevel design and managers’ self-assessments [51].

This empirical study demonstrates that teleworkers who engage in presenteeism and perceive their managers’ stress management competencies positively experience better mental health and job performance [29], steering them towards more functional presenteeism. The connection is even stronger when employees either agree with their managers’ high stress management competencies or overestimate them compared to the managers’ own self-assessments.

This empirical study highlights that managers’ stress management competencies perceived by teleworkers who engage in presenteeism have a positive relation with their mental health and job performance, guiding them towards a more functional presenteeism. The link is even stronger if they agree strongly with their managers or if they overestimate their managers’ competencies.

Interestingly, this study did not find a relationship between managers’ self-assessment and the mental health or job performance of teleworkers engaged in presenteeism [47]. However, a relationship was observed when managers actively sought to supervise and communicate about both current and future work. This finding highlights the influence of communication within such a context. The disparity in results between employees’ perceptions and managers’ self-assessment implies that managers and employees are likely to engage in diverse interactions and assessments of managers’ behavior, leading to disparate information and weaker convergence in assessments [44]. As noted in the latest Toderi and al. [51] research, the correlations between managers’ and employees’ ratings of the four management competency behaviors were all non-significant. This indicates considerable variation between self-ratings and others’ ratings, highlighting the need for perceptual congruence analyses.

Our findings also support the notion that manager self-assessment and employee-perceived SMCs are positively related, albeit weakly, as argued by Jacobsen and Bogh Andersen [46]. This is what we see in face-to-face studies and, therefore, probably exacerbated by telework, considering that teleworkers have fewer opportunities to observe management behaviors. H4a received support, suggesting that a strong agreement between managers and employees with a high assessment perception of competencies is linked to increased functional presenteeism. Even when employees perceived their managers’ competencies more positively than their managers themselves (H4d), there was a positive effect on functional presenteeism, although it was less pronounced. Conversely, low assessment agreement (H4b), or when employees underestimate their managers’ competencies (H4c), had a lower positive association with mental health and job performance. 

In summary, the two profiles that most effectively reduce psychological distress and improve job performance are those where employees have a positive perception of their manager, regardless of the managers’ self-assessment. This suggests that, to promote functional presenteeism (better mental health and job performance), it is beneficial for employees to perceive their managers as having strong stress management competencies, as these are viewed as valuable resources. These results suggest that what matters is employees’ positive perception of their managers’ SMCs. It suggests we need to put more attention on increasing positive employee perceptions of managers’ stress management competencies to increase functional presenteeism among teleworkers. To improve employees’ perceptions of managers’ stress management competencies and boost functional presenteeism, invest in manager training, establish feedback channels, promote open communication, recognize effective stress management, provide necessary resources, encourage modelling of positive behaviors, conduct regular check-ins, and offer personalized support based on individual needs.

### 4.1. Theoretical Contributions

This study contributes to the field of occupational health psychology in several key ways. Firstly, this study addressed a research gap by investigating the management of virtual presenteeism, an area that has not previously been explored. Secondly, by examining how managers’ stress management competencies (SMCs) facilitate higher mental health and job performance, which is linked to more functional presenteeism during telework, it provides empirical support for the HPFP Framework [12].

Thirdly, the study enhances our understanding of the relationship between SMCs and employees’ mental health and job performance in the context of virtual presenteeism by combining a multilevel conceptual model with the SMC reference framework (i.e., competencies perceived by employees and managers’ self-assessment) [30]. This novel approach offers insights into managing virtual presenteeism effectively and establishes theoretical foundations for future research in this area.

Furthermore, this study enriches the literature on the Self-Other Agreement model [43,44] by using a multilevel design to examine perceptual gaps between employees and managers, incorporating multisource assessment. This provides valuable insights for practitioners and managers into developing SMCs. Finally, the findings illuminate the skills and competencies necessary for effective telework management, an area that has received limited attention in research [30].

### 4.2. Practical Implications

The practical implications of this study support initiatives to enhance virtual presenteeism management skills, aiming to improve mental well-being and overall workplace performance. Human resource managers should therefore promote the development of these competencies within their organizations to mitigate the negative effects of presenteeism on employees’ mental health and job performance [12]. 

Firstly, the SMC framework [30] is a valuable model for managers dealing with virtual presenteeism. Managers are advised to demonstrate strong interpersonal skills by being compassionate, managing emotions effectively, and upholding integrity. They should also adopt proactive work management practices, such as using participative and problem-solving approaches. Additionally, managers should proactively address conflicts, maintain open communication, and seek mutually beneficial solutions to cultivate a positive work environment. Efficient use of organizational resources is crucial to conflict management, involving prompt resolution while seeking support from human resources, conflict resolution specialists, or mediation services when needed. Finally, these competencies can reach their full potential when managers are approachable, sociable, and demonstrate empathetic engagement. These qualities can help foster a supportive and inclusive work environment, ultimately benefiting both employees and the organization.

The absence of clear communication cues in remote work can prevent managers from identifying presenteeism [20] or employees from recognizing their managers’ competencies, and identifying distressed employees and maintaining their mental health and performance during remote work can be challenging [56]. The minimal relationship between managers’ self-assessments and employees’ perceptions of managers’ SMCs suggests either that managers are not actively engaging in stress management competencies or that their efforts are not significantly related to employees’ mental health and performance, potentially remaining unrecognized by employees. Thus, organizations must be mindful of the potential risks associated with telecommuting [7], and managers must understand that their stress management efforts may go unnoticed due to the isolation of remote work. Increasing awareness of manager behaviors and providing training may positively impact employee mental health and job performance. Managers must demonstrate their competencies more effectively in teleworking environments to ensure their positive self-perceptions are conveyed to employees.

Ensuring a positive perception requires effective communication in terms of both frequency and quality [57], and actively seeking feedback from employees [58]. However, remote work reduces opportunities for seeking or receiving informal performance feedback [59] and for recognizing employees’ skills and contributions [58], which makes consistent evaluation challenging.

Additionally, given the heightened importance of communication in remote work and the potential exaggeration of employee perception biases, specific training in managing virtual presenteeism may be beneficial. Workshops focusing on this area could help managers better understand the significance of their behaviors, enhance their self-awareness, and empower them to foster positive competencies [30].

### 4.3. Limitations and Future Directions

Despite the study’s strengths, especially the use of multilevel analysis that incorporated assessments from multiple data sources (e.g., levels of agreement between managers and employees), some limitations must be considered when interpreting our findings. The first issue is related to the small number of manager-employee clusters in our study (157 clusters, averaging 4.6 cases per cluster), which reduces its statistical power. However, researchers suggest having at least 50–100 clusters [60], or simpler models with 50 or more groups of 5–10 cases that are less likely to face convergence problems (e.g., [61]). Despite the limited statistical power, we were still able to identify several significant interactions that are of interest to future study.

It is important to highlight that the study’s Cronbach’s alpha values align with the findings of Toderi and Sarchielli [41]. However, it is noteworthy that the Respectful and Responsible (RR) subscale for the manager self-assessment showed significantly lower internal consistency, with a Cronbach’s alpha of α = 0.35. This discrepancy suggests that the RR subscale items may not be as relevant or well understood by the managers in our sample, leading to greater variability in their responses. Future research should consider reviewing these items to ensure their clarity and contextual relevance.

The study used an 80% threshold to classify employees as full-time teleworkers. This approach may introduce a limitation, as it could lead to an overestimation of presenteeism among teleworkers if health problems were specifically experienced during the 20% of time spent in the office rather than during telework. Future research should address this limitation by clearly defining the telework period and specifically asking participants to reflect on presenteeism experienced during their telework period. Additionally, we measured presenteeism and telework using two different recall periods (seven days for telework and one month for presenteeism), which is a notable limitation. This approach introduces a potential risk that illness days may fall outside of telework days. Future research should address this issue by aligning the recall periods for both telework and presenteeism to ensure consistency.

The study also relied on self-report survey measures of presenteeism, which may introduce some degree of common method bias [62]. The inherent subjectivity of presenteeism suggests that occurrences are inevitably self-reported, which is customary in this line of research. Thus, following other presenteeism research, we depended on participants’ subjective assessments of whether their health status justified taking time off work, which precluded objective assessment. Furthermore, when completing the PD and JP scales, participants were not asked about specific days but about the past month, which raises questions about the most suitable method for measuring presenteeism. Furthermore, this research did not differentiate between presenteeism due to mental or physical health issues, which could be considered a limitation since our focus was solely on the link with mental health. Some might argue that it would have been more consistent to select only employees with mental health issues. Moreover, we included all instances of presenteeism, regardless of the underlying cause of illness, due in part to the sample size of our participants. Future research might benefit from specifically examining virtual presenteeism related to mental health issues for greater coherence in studying the connection with employee mental health.

Another potential limitation is the use of a self-assessment job performance too, highlighting the limitations of self-assessment questionnaires compared to objective measures, particularly regarding reliability and potential biases such as leniency and social desirability [63]. However, the decision to use the IWPQ, a self-report questionnaire, was intentional. Objective performance measures are often impractical, especially in knowledge work or high-complexity jobs where quantifiable outcomes like production quantity or error rates are not feasible [64]. Self-reports, on the other hand, allow individuals to reflect on their own experiences, making them well suited to assess personal outcomes such as mental health and job performance. Our study aimed to examine the relationship of stress management competencies (SMCs) with employees’ subjective experiences of mental health and job performance in a context of virtual presenteeism. While managerial ratings might be affected by the halo effect [39,65], self-reports offer practical benefits, including ease of data collection, confidentiality, and reduced issues with missing data [66]. In situations where objective measures are challenging or unattainable, self-reports provide a viable alternative for evaluating performance [67]. The focus of our study was to capture the factors influencing employees’ subjective evaluations of their own behaviors and environments. 

Finally, the generalizability of our research may be limited for three reasons: First, the study focuse on a single, large, public organization in Québec. Second, the field of activity may be seen as a limitation of the study as the organization operates in a unique public sector framework with a specific mandate, thus, the findings related to employee and manager behavior, presenteeism, and well-being might not be easily generalizable to other sectors or organizations, particularly those in the private sector or outside regulatory frameworks. Third, the study was conducted in 2022 during the COVID-19 pandemic, when the government imposed quarantine/lockdown measures. Consequently, replications of our study conducted over time, in different countries, and in the public and private sectors would bolster our confidence in the broader applicability of the results.

Future studies should consider the organizational contextual factors that promote better recognition of perceptual differences between managers and employees, and explore how these factors could improve SMC adoption among managers to support employees’ mental health and job performance. Future research may also consider how managers might detect employees engaged in virtual presenteeism and monitor their mental health without infringing on privacy rights [68]. Finally, employee characteristics may also influence how people perceive leadership behaviors. While sociodemographic characteristics play a minor role in the results, personality traits significantly affect how followers rate their managers [69]. Ultimately, our results reveal new research avenues concerning the misalignment between manager self-assessments and employee perceptions of SMCs.

## 5. Conclusions

Our study makes significant progress in understanding functional virtual presenteeism by providing the first direct empirical evidence linking stress management competencies (SMCs) with improved employee mental health and job performance. Our findings indicate that employees’ perceptions of their managers’ SMCs are crucial in shaping their own mental health and job performance. This impact is even more pronounced when employees have a high perception of their managers’ SMCs, whether they overestimate or strongly agree with their managers’ self-assessments. Additionally, our study reveals a notable gap between managers’ self-assessments and employees’ perceptions of these competencies. By emphasizing the importance of high alignment on managers’ SMCs and fostering positive employee perceptions, our research underscores the critical role of employees’ positive assessments of their managers’ stress management competencies. Ideally, managers can work to improve employee perceptions by actively demonstrating all SMCs, enhancing communication, and providing regular feedback to close perception gaps and promote functional presenteeism, which translates into better mental health and job performance in the workplace.

## Figures and Tables

**Table 1 ijerph-21-01115-t001:** Framework of managers’ and employees’ stress management competence (SMC) perception gap assessments.

(−) Manager Competency Self-Assessment (+)	Disagreement- Underestimate (DU)	Agreement High (AH)
	Agreement Low (AL)	Disagreement- Overestimate (DO)
	(−) Employee Competency Assessment (+)	

**Table 2 ijerph-21-01115-t002:** Sociodemographic data.

	Managers					Employees				
Descriptive Statistics	N	Min	Max	Mean	SD	N	Min	Max	Mean	SD
Gender	157	1.00	2.00	1.39	0.490	360	1.00	2.00	1.20	0.398
Age	157	1.00	3.00	2.01	0.664	364	2.00	4.00	2.69	0.750
Education level	100	1.00	3.00	2.26	0.597	364	1.00	4.00	2.75	0.787
Job level	100	1.00	3.00	1.37	0.691	364	1.00	3.00	1.35	0.485

Note: Level 1 (managers; N = 157); Level 2 (employees; N = 365). Gender codes: 0 = male, 1 = female. Age codes: 1 = <25, 2 = 25–44, 3 = 45–54, 4 = ≥55. Education-level codes: 1 = high school, 2 = college, 3 = bachelor’s degree, 4 = master’s or doctorate. Job-level codes: 1 = professional, 2 = civil servants and workers. CI = 95% confidence interval

**Table 3 ijerph-21-01115-t003:** Subscale intraclass correlation coefficients (ICC).

Variable	Residual	Estimate	Std. Error	Wald Z	Sig.
Psychological distress (PD)	16.59	2.02	1.07	1.89	0.06
Job performance (JP)	0.31	0.03	0.02	1.43	0.15

**Table 4 ijerph-21-01115-t004:** Descriptive statistics and study variable correlations.

Descriptive Statistics	Mean	SD	1	2	3	4	5	6	7	8	9	10
1. RR (M)	3.88	0.28	—									
2. MCW (M)	4.13	0.32	0.44 **	—								
3. RDS (M)	4.19	0.40	0.47 **	0.59 **	—							
4. MIT (M)	4.39	0.31	0.32 **	0.45 **	0.63 **	—						
5. RR (E)	4.09	0.59	0.10	−0.00	0.02	0.03	—					
6. MCW (E)	3.83	0.70	0.02	0.09	0.00	0.07	0.63 **	—				
7. RDS (E)	3.93	0.78	0.03	0.04	0.06	0.05	0.56 **	0.67 **	—			
8. MIT (E)	4.07	0.67	0.02	0.04	0.01	0.05	0.59 **	0.71 **	0.63 **	—		
9. PD (E)	7.75	4.37	−0.01	−0.10	−0.09	−0.07	−0.32 **	−0.33 **	−0.18 **	−0.21 **	—	
10. JP (E)	3.53	0.58	0.05	0.12 *	−0.01	0.01	0.26 **	0.36 **	0.18 **	0.19 **	−0.46 **	—

Note: E = Teleworking employee who reports presenteeism, M = manager, RR = respectful and responsible, MCW = managing and communicating existing and future work, RDS = reasoning and managing difficult situations, MIT = managing the individuals within the team. PD = psychological distress, JP = job performance. Level 1 (M; N = 157); Level 2 (E; N = 365). * *p* < 0.05, ** *p* < 0.01.

**Table 5 ijerph-21-01115-t005:** Multilevel correlations between managers’ self-reported SMCs and employees’ perceptions of their manager’s SMCs, and virtual presentees’ mental health and job performance.

SMCs (M)	PD	JP	SMCs (E)	PD	JP
RR	0.00	0.05	RR	−0.33 **	0.26 **
MCW	−0.10	0.12 **	MCW	−0.31 **	0.34 **
RDS	−0.09	0.00	RDS	−0.17 **	0.18 **
MIT	−0.06	0.01	MIT	−0.21 **	0.18 **

Note: E = Teleworking employee who reports presenteeism, M = Manager, RR = respectful and responsible, MCW = managing and communicating existing and future work, RDS = reasoning and managing difficult situations, MIT = managing the individuals within the team, PD = psychological distress, JP = job performance. Level 1 (M; N = 157); Level 2 (E; N = 365). CI = 95% confidence interval. ** *p* < 0.05.

**Table 6 ijerph-21-01115-t006:** Multilevel intercorrelations between managers’ self-reported SMCs and employees’ perceptions of their managers’ SMCs.

SMCs	RR (E)	MCW (E)	RDS (E)	MIT (E)
RR (M)	0.08	0.00	0.02	0.02
MCW (M)	0.01	0.10 **τ**	0.04	0.04
RDS (M)	0.02	0.01	0.06	0.01
MIT (M)	0.04	0.09	0.05	0.06

Note: E = Teleworking employee who reports presenteeism, M = manager, RR = respectful and responsible, MCW = managing and communicating existing and future work, RDS = reasoning and managing difficult situations, MIT = managing the individuals within the team. Level 1 (M; N = 157); Level 2 (E; N = 365). **τ**
*p* < 0.10.

**Table 7 ijerph-21-01115-t007:** Employee-manager agreement on manager’s SMCs and association with virtual presentees’ mental health and job performance.

	Psychological Distress				Job Performance			
	Quadrant	Estimate	t Value	*p* Value	Quadrant	Estimate	t Value	*p* Value
**RR**	*Agree-Low*	8.7987 _a_	18.66	<0.0001	*Agree-High*	3.6474 _a_	60.94	<0.0001
	*Disagree-Under*	8.6838 _a_	17.64	<0.0001	*Disagree-Over*	3.6312 _ab_	57.53	<0.0001
	*Disagree-Over*	6.8161 _b_	14.01	<0.0001	*Disagree-Under*	3.4556 _bc_	53.72	<0.0001
	*Agree-High*	6.7872 _b_	14.66	<0.0001	*Agree-Low*	3.3655 _c_	55.11	<0.0001
**MCW**	*Agree-Low*	8.9434 _a_	19.89	<0.0001	*Agree-High*	3.7941 _a_	63.65	<0.0001
	*Disagree-Under*	8.8036 _a_	18.83	<0.0001	*Disagree-Over*	3.6518 _a_	59.05	<0.0001
	*Disagree-Over*	7.0256 _b_	14.37	<0.0001	*Disagree-Under*	3.4014 _b_	57.52	<0.0001
	*Agree-High*	5.9873 _b_	12.61	<0.0001	*Agree-Low*	3.2990 _b_	58.39	<0.0001
**RDS**	*Disagree-Under*	8.0552 _a_	19.06	<0.0001	*Agree-High*	3.6662 _a_	49.80	<0.0001
	*Agree-Low*	7.7633 _a_	18.11	<0.0001	*Disagree-Over*	3.6046 _ab_	42.98	<0.0001
	*Disagree-Over*	7.6559 _a_	11.95	<0.0001	*Disagree-Under*	3.4758 _b_	64.55	<0.0001
	*Agree-High*	7.2340 _a_	12.71	<0.0001	*Agree-Low*	3.4735 _b_	64.33	<0.0001
**MIT**	*Agree-Low*	9.0806 _a_	18.86	<0.0001	*Disagree-Over*	3.7030 _a_	55.31	<0.0001
	*Disagree-Under*	7.8798 _ba_	17.76	<0.0001	*Agree-High*	3.6312 _a_	60.55	<0.0001
	*Agree-High*	7.0800 _b_	15.08	<0.0001	*Disagree-Under*	3.4622 _b_	61.25	<0.0001
	*Disagree-Over*	6.8132 _b_	12.96	<0.0001	*Agree-Low*	3.3436 _b_	54.05	<0.0001

Note: Subscripts with different letters are significantly different; RR = respectful and responsible; MCW = managing and communicating existing and future work; RDS = reasoning and managing difficult situations; MIT = managing the individual within the team.

## Data Availability

The data that support the findings of this study are available from the corresponding author, (S.S.) upon reasonable request.

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
