# Peer review of "Managing Virtual Presenteeism during the COVID-19 Pandemic: A Multilevel Study on Managers’ Stress Management Competencies to Foster Functional Presenteeism"

_ijerph, 2024, doi:10.3390/ijerph21091115_

Round 1

Reviewer 1 Report

Comments and Suggestions for Authors

Dear Authors. It has been very interesting to read your manuscript. 

My main concern relates to the aim and focus. 

When reading the abstract and introduction I see different foci. 

is it about how  SMC influence MH/Performance? L15

or about the relation between SMC and presenteeism? and is it about all 4 types of presenteeism or just one?

or is it about SMC in the context of teleworking? And who's perception of SMC is in focus and what is the reason for exploring the agreement of the managers(?) perception. Who has a problem in practice L12?

or is it the agreement (or not) that affects the Mental/performance or presenteeism? 

and finally: are you concerned about people teleworking when ill (so any illness) or mental health issues/problems?  

from the methods section, I understand that people were strongly recommended to work from home or it was mandatory. So, how do managers gain information and spot people working while ill? That is my understanding of the paper. But when reading it, you also indicate that mental health problems are linked to managers' SMCs.

check L. 425 - how can SMCs foster functional presenteeism? if that is the focus that should be stated up front, and the other aims should be "cleaned up". or is the aim, in fact: Does managers' SMC (as perceived by the employees teleworking) foster functional presenteeism?  

Please look into the motive and aim of the study. That will also help in understanding the conclusions: 

I have some comments that I hope you will look into. 

1. Your title has two messages  - is your work about virtual presenteeism? of about employee-manager agreement on Stress management competencies? 

2. Line 14. whos competencies do you focus on? Please clarify. 

L.20 - I am in doubt if you focus on experiences or perceptions when I read your MS. please clarify

L23 - who's perception of managers' or employees' perception of managers?

L36 - when is now? please specify

L 44 - employees or workers - you mix in the MS. 

49-51: difficult to read. 

from line 50. I miss the opposite position - that people recover when WFH - where WFH allows for some work  - the current perspective does not have a preventive or promotion focus. 

L80 - a . is missing before It.

The flow in the MS could be improved. you could consider switching the sections 1.1 and 1.2 

L104 - the presentation of the HPFP is a bit abrupt. You could consider to state/mention that there are 4 different presenteeism profiles and that neither of these have been considered in a virtual set up. 

L124 not sure of the link/logic to the paragraph above and below - seems to be wedged in

L.131 - maybe this paragraph should be placed before L103? 

L150 - who's presenteeism management? the managers?

L151 - 163 - suggest knowledge is missing.  you could consider moving this part up as the motivation for the study could be clearer.

L 184 - which period? lost lockdown? 

H1a/b - who's competencies do you focus on? the employees' or their managers?

L 221 - how do you assess weakly? and from which perspective? is it weakly positive or negative?  see also L 445 - where you write that it is positive

L 225 - little is known about linke between self-assesment and employee perceptions or is it virtual preseenteeism? or both?   

L. 239 + : agreement - when is that reached? and among all/ the whole population or one manager's own employees ? What is meant by underestimate? do you mean misalignment/ or disagreement? how is the underestimate defined? 

L. 287 - you asked about health issues - how did you separate the answers related to mental health? ok - I get it - when I get to L314 - you could consider to combine health and mental health

and maybe move up the telework section as that is your key selection criteria  - L292

L 354 - employee perceptions of ? 

L355 - which 2 perspectives? too implicit what you mean

L358 - this section seems to belong in the discussion - you could consider moving it

L370 - please spell out PD - 

L 375 - what is MCW SMCs  - difficult to read/ follow with all the abbreviations  - please write out the claims to make following the analysis. the presentation of the results becomes very staccato and the learnings from the analysis are left to reader to do. 

L 434 - if that is your claim the motivation and aim should be clarified. 

L. 458 - you could consider to rewrite the sentence. ...matters is employees' positive perception of their manager's SCM. 

L461 - where do you write about the need of enhancing effective managerial behaviour? in the introduction it is more about general managerial challenges. 

L 463 - I suggest that you focus on functional preseentism already in the introduction as it "falls from the sky" during the manuscript that it is your focus in this paper. 

What competencies do the managers have and exercise in your dataset - What do you find? that is not clear, and therefore, I miss a discussion of what you have found is similar to what the literature presents or differs from.

L. 500. I agree that managers' SM activities may go unnoticed- but did your findings show that ? or what is the link to your study? do you have results showing that managers make an effort but it is not noticed by their employees? 

the end of the discussion is more about managers being able to communicate their SM practices and showing their competencies, and how to train them to manage virtual presenteeism, but you write nothing about functional presenteeism. 

L. 556+ - difficult to read - what is the exact point you aim to make? 

the last line - is it so that if the employees have a positive perception of their manager's SCM then their mental health will be good? so it is all about perception? as mentioned in the beginning of my comments, there is a need to clarify the purpose of the study and the focus. As you can probably see from my comments, it is not all clear to me what it is you aim to study and can claim. 

Reviewer 2 Report

Comments and Suggestions for Authors

Introduction:

-          Topic is very interesting and less presented in existing literature

-          Provide details of the field / sector of activity of the “large public sector organization” & details of the field of activity of the employees; any specific sectors / any specific jobs? à is this another limitation of the study

-          What does “multilevel data” mean? (line 266)

-          Is there a mistaken in Hypothesis H4b? [“H4b: Low manager–employee agreement about stress management competencies assessment is associated with lower psychological distress and higher job performance” - > the same as H4A and inconsistent with the results (see lines 417-418) “Agree-Low assessments will be detrimental to JP and increase PD”

Materials and Methods

-          It is mentioned that “Employees who worked remotely for at least 80% of the time in the previous 28 day” (line 263) and then “Participants were asked to indicate the number of hours worked from home and the number of hours worked onsite during the past week (seven days).” (lines 293-294) … there is a contradiction here

-          Is there a risk that the employee experienced health problem during office work (20%) and not telework? Again, not sure if the 20% of office work comes from “last week” or “last 28 days”

-          It is mentioned that “Employees who declared at least one day of presenteeism during the 28-day period were considered to have engaged in virtual presenteeism (N = 365)” (lines 263-264) -> but the data referring to telework are consider in the “last week” è what if the day of illness was during office days, or outside the last seven days?

Job Performance:

-          The “Individual Work Performance Questionnaire” is a self-assessed performance? Are there any limitation in this? see Discussion section> “Although we identified a notable disparity between manager self-assessments and employee perceptions of SMCs, it is the employee’s personal evaluation, not the manager’s self-assessment, that is the most influential”.´Wwhat if the same happens when assessing job performance  (self-assessment vs managers’ assessment ->  in this case is the managers’ assessment more influential? (the one not taken into consideration in this study)

-          what period is it taking into consideration for assessing job performance? Is it the same 28 last days, to be in line with the “mental health assessment” or is it “in general”? – the same question for SMCIT-36
